# Effects of a Long-Term Wearable Activity Tracker-Based Exercise Intervention on Cardiac Morphology and Function of Patients with Cystic Fibrosis

**DOI:** 10.3390/s22134884

**Published:** 2022-06-28

**Authors:** Maria Anifanti, Stavros Giannakoulakos, Elpis Hatziagorou, Asterios Kampouras, John Tsanakas, Asterios Deligiannis, Evangelia Kouidi

**Affiliations:** 1Laboratory of Sports Medicine, Aristotle University of Thessaloniki, 57001 Thermi, Greece; manyfant@phed.auth.gr (M.A.); stgiannakoulakos@gmail.com (S.G.); adeligia@phed.auth.gr (A.D.); 2Pediatric Pulmonology and Cystic Fibrosis Unit, Hippokration Hospital, Aristotle University of Thessaloniki, 54642 Thessaloniki, Greece; ehatziagorou@gmail.com (E.H.); asterioskampouras@gmail.com (A.K.); tsanakasj@gmail.com (J.T.)

**Keywords:** cystic fibrosis, exercise training, wearable activity trackers, 6-min walking test, echocardiography

## Abstract

Several studies have shown that patients with cystic fibrosis (CF), even at a young age, have pulmonary and cardiac abnormalities. The main complications are cardiac right ventricular (RV) systolic and/or diastolic dysfunction and pulmonary hypertension, which affects their prognosis. Exercise training (ET) is recommended in patients with CF as a therapeutic modality to improve physical fitness and health-related quality of life. However, questions remain regarding its optimal effective and safe dose and its effects on the patients’ cardiac function. The study aimed to provide a wearable activity tracker (WAT)-based ET to promote physical activity in CF patients and assess its effects on cardiac morphology and function. Forty-two stable CF individuals (aged 16.8 ± 3.6 years) were randomly assigned to either the intervention (Group A) or the control group (Group B). Group A participated in a 1-year WAT-based ET program three times per week. All patients underwent a 6-min walking test (6-MWT) and an echocardiographic assessment focused mainly on RV anatomy and function at the baseline and the end of the study. RV systolic function was evaluated by measuring the tricuspid annular plane systolic excursion (TAPSE), the systolic tricuspid annular velocity (TVS’), the RV free-wall longitudinal strain (RVFWSL), and the right ventricular four-chamber longitudinal strain (RV4CSL). RV diastolic function was assessed using early (TVE) and late (TVA) diastolic transtricuspid flow velocity and their ratio TVE/A. Pulmonary artery systolic pressure (PASP) was also estimated. In Group A after ET, the 6MWT distance improved by 20.6% (*p* < 0.05), TVA decreased by 17% (*p* < 0.05), and TVE/A increased by 13.2% (*p* < 0.05). Moreover, TAPSE, TVS’, RVFWSL, and RV4CSL increased by 8.3% (*p* < 0.05), 9.0% (*p* < 0.05), 13.7% (*p* < 0.05), and 26.7% (*p* < 0.05), respectively, while PASP decreased by 7.6% (*p* < 0.05). At the end of the study, there was a significant linear correlation between the number of steps and the PASP (r = −0.727, *p* < 0.01) as well as the indices of RV systolic function in Group A. In conclusion, WAT is a valuable tool for implementing an effective ET program in CF. Furthermore, ET has a positive effect on RV systolic and diastolic function.

## 1. Introduction

Patients with cystic fibrosis (CF) often showed pulmonary and cardiovascular complications such as pulmonary hypertension, etc. and perform less physical activity than their healthy counterparts [1,2,3]. Some of the main reasons include respiratory insufficiency due to pathological remodeling of the lung, inspiratory muscle weakness, cardiac dysfunction in severe cases, and peripheral muscle dysfunction [1,2]. In addition, their sedentary behavior is an essential factor affecting disease progression, morbidity, and mortality [2]. Therefore, in recent years, there have been many efforts by social and scientific bodies to convince patients with CF to follow a physically active lifestyle. Thus, studies have highlighted the health benefits of regular physical activity in CF. However, although many studies have reported the significant impact of exercise training (ET) on physical fitness, quality of life, and respiratory function, no studies have examined the effects of exercise training on the patients’ cardiac function.

Many questions remain about how ET can affect the CF patients’ respiratory and cardiovascular systems. Moreover, uncertain results exist concerning the type of exercise that is the most appropriate for these patients and the most effective and safe ET prescription in terms of intensity, duration, and frequency [3,4]. Consumer wearable activity tracker (WAT)-based interventions have been used in sports for self-monitoring ET performance-related parameters [5]. Their use in digital health, specifically mobile-rehabilitation programs for patients with chronic diseases, has recently been extended [6]. However, the clinometric properties of the objective and subjective physical capacity assessment tools (reliability, validity, and responsiveness) have not been sufficiently evaluated in CF patients [7]. Few studies have used motion sensors to monitor physical activity dimensions such as energy expenditure, the number of steps, and time spent at different intensities including sedentary time in CF, but have not described definite results [8,9]

Thus, the present study aimed to use WATs to adequately prescribe the scheduled ET program dose applied to patients with CF and investigate whether it was sufficient to cause improvements in physical function and cardiac structure and function.

## 2. Materials and Methods

### 2.1. Study Population

Patients with CF were recruited from the Pediatric Pulmonology and Cystic Fibrosis Unit, Hippokration Hospital, Aristotle University of Thessaloniki, Greece.

The study’s inclusion criteria were the following: patients with a stable clinical status, without worsening respiratory symptoms or overt pulmonary hypertension, or a history of contemporary heart issues. In addition, patients already participating in structured physical activity were excluded from the study. Finally, out of the 75 patients tested for eligibility for the study, nine eventually refused to participate (seven due to long distance stays and both due to occupational employment) and the remaining 16 due to the presence of non-eligible criteria: active manifestation respiratory infection (six patients), findings of pulmonary hypertension (four patients), co-existing cardiac problems (two patients), and their regular participation in regular exercise programs (four patients)

Patients fulfilling the inclusion criteria were informed regarding the aim and methodology of the research protocol and provided informed consent. Moreover, in the cases of minors, the parents/guardians signed the consent form. The Ethics Committee of the Aristotle University of Thessaloniki approved the study protocol. The study was conducted in accordance with the Declaration of Helsinki.

### 2.2. Study Protocol

The eligible patients’ anthropometric characteristics (weight, height, BSA, BMI), clinical, medical, and physical activity history, and current medications were collected. After that, all patients underwent a six-minute walking test and an echocardiographic assessment, repeated at the end of the 1-year study. All outcome assessors were blinded to patient group allocation. After the initial assessment, the eligible patients were randomly assigned in a 1:1 ratio by simple randomization (drawing lots) to either the intervention (Group A) or the control group (Group B). The exercise group participated in a 1-year WAT-based ET program three times per week. Patients in the control group were asked to continue their usual lifestyle to participate in recreational physical activities, but refrain from any structured exercise intervention.

### 2.3. Measurements

#### 2.3.1. Functional Capacity Assessment

A 6-min walking test (6MWT) was performed in both groups to evaluate the individuals’ level of physical fitness. This test measured the distance (6MWTD) that each patient could quickly walk on a flat and hard surface (30 m) in 6 min. In addition, the Borg scale Rating of Perceived Exertion Exercise was used to monitor the perceived intensity during the workouts.

#### 2.3.2. Echocardiographic Study

Echocardiographic studies were performed using a Vivid S70; GE Medical; Horten, Norway, equipped with an M5S phased-array transducer. Each subject was examined in a semisupine left lateral position. The electrocardiogram was recorded continuously, and three consecutive beats were stored digitally. All studies were stored in a central workstation (Echopac, version 201) and were analyzed offline. The internal dimension of the left ventricle (LVEDD), the left ventricular interventricular septum thickness (LVIVSd), and the posterior wall thickness (LVPWd) were obtained at the end-diastole from the parasternal long-axis view. The dimension of the left atrium (LA) was measured at the end-systole. The systolic function of the left ventricle (LV) was acquired from the apical 4- and 2-chamber views. The LV end diastolic volume (LVEDV) and LV ejection fraction (LVEF) were obtained by the modified biplane Simpson’s method. The assessment of diastolic function was estimated with the measurement of early diastolic transmitral flow velocity (MVE), late diastolic transmitral flow velocity (MVA), and their ratio (MVE/A). All measurements were undertaken following the European Association of Cardiovascular Imaging and American Society of Echocardiography guidelines [10].

Right ventricular (RV) linear dimensions were assessed from parasternal long axis proximal RV outflow diameter (RVOT prox) and RV-focused view basal RV linear dimension (RV bas) at the end-diastole. Right atrial volume (RAVol) was calculated using the disk summation techniques from apical four-chamber view at end-systole. The diameters of RV and RAVol were normalized with body surface area. The measurement of tricuspid annular plane systolic excursion (TAPSE) was obtained by M-mode echocardiography with the cursor optimally aligned along the direction of the lateral tricuspid annulus apical four-chamber view. The diastolic function of RV was measured from early diastolic transtricuspid flow velocity (TVE), late diastolic transtricuspid flow velocity (TVA), and their ratio (TVE/A) at held end-expiration without tachycardia. Pulsed-wave TDI analysis of the RV free wall at the level of the tricuspid valve annulus was performed to measure the systolic tricuspid annular velocity (TVS’), the early diastolic tricuspid annular velocity (TV), and the late diastolic tricuspid annular velocity (TVA’). Pulmonary artery systolic pressure (PASP) was measured from peak TR jet velocity, using the simplified Bernoulli equation and combining this value with an estimate of the RA pressure, estimated from the IVC diameter and respiratory changes following the guidelines. The 2D systolic longitudinal strain was measured from an RV-focused apical view, and the RV was divided into six segments (basal, mid, and apical segments of the RV free wall and interventricular septum). The RV free-wall longitudinal strain (RVFWSL) and the RV four-chamber longitudinal strain (RV4CSL) were measured following the consensus document of the EACVI/ASE Industry [11].

#### 2.3.3. WAT-Based ET Program

The 1-year intervention consisted of an individualized home-based ET program using WAT to ensure the patients’ autonomy and often behavioral face-to-face sessions with a physician. The exercise program consisted of any physical activity that each patient preferred that counted steps at least five times a week. The goal was for females to achieve >10,000 steps and males >12,000 per day during the third month of the program and gradually increase to >14,000 steps for females and >16,000 steps for males toward the end of the 1-year intervention. To avoid injuries during a scheduled ET session, patients were asked to start with 5 min warm-up and stretching exercises and finish with 5-min cool down. Patients were asked to choose enjoyable, easy, and accessible activities such as brisk walking, jogging, or dancing, at least five days a week. They were also encouraged to participate in any recreational sports they preferred to increase adherence. Τheir steps during these activities, where the sport allowed it, were counted in the number of total steps. Each patient was asked to upload and store their WAT data on the computer by the end of each week for 52 weeks. The same researcher collected the data, monitored each patient’s progress, and called them in case they did not achieve the goal. The face-to-face behavioral sessions with a physician occurred during the clinic visits at the baseline and every three months. At all sessions, patients brought their diaries, presented their progress, described their experiences and impressions, discussed their achievements, and were educated about the health benefits of exercise, encouraged to continue the ET program, and instructed to continue the self-monitoring to achieve their goal. Therefore, patients were asked to note the number of steps during their daily physical activity and not its duration. Moreover, the total step counts were computed by summing each exercise training step and calculating the median values from the ET days for each participant. At the end of the study, the sum of the total steps per week was estimated. Regular clinical examinations were also performed in all face-to-face clinical sessions.

The G1 (Garmin, Schaffhausen, Switzerland) wearable activity tracker, a 3-axis accelerometer, was provided to each patient in Group A to record their level of physical activity in terms of the number of steps per day. Participants were asked to wear the WAT continuously during the study. They were also instructed to record the number of steps achieved each day in their diary. Additionally, they were educated to upload their data online weekly at http://www.garminconnect.com (accessed on 23 March 2022).

To assess the effective behavioral principles (acceptance of the WAT, goal setting, self-monitoring, personalized feedback), each patient filled out a simple questionnaire at the end of the study, answering the following questions:Did you accept wearing the wearable activity tracker sensor during your exercise training? (Yes/No).Do you think that the wearable activity tracker provided you support and motivation to increase your level of participation in the exercise training program? (1 = not at all/10 = very much).Do you think the wearable activity tracker increased your adherence to the exercise training program? (1 = not at all/10 = very much).

### 2.4. Statistical Analysis

Data are presented as mean ± SD. The Statistical Package for Social Sciences (SPSS, Chicago, IL, USA), version 25.0 software for windows, was used. The normality of distribution was examined using the Kolmogorov–Smirnov test T-test was used to examine the initial mean differences between the two groups, while two-way repeated measures ANOVA was used to examine the mean differences within time and between the two groups with a Bonferroni correction (Bonferroni post hoc test). Moreover, linear regression was used to study the association between variables at the baseline and the end of the study. The two-tailed *p*-values < 0.05 were considered as statistically significant.

## 3. Results

Fifty patients with CF participated in the study (23 males, aged 16.8 ± 3.6 years). Eight participants (four from each group) dropped out of the study due to exacerbation or non-compliance (Figure 1). All participants in Group A, except one, fulfilled at least 80% of all training sessions and were analyzed. No adverse effect from the ET program was observed.

The demographic and clinical characteristics of the 42 patients (21 in each group) at baseline and after one year are presented in Table 1.

In Group A, the mean number of steps per day recorded by the WAT was at baseline 6374.4 ± 812.1 and 7543.2 ± 654.1; at the end of six months, 10,103.3 ± 985.9 and 12,464.8 ± 846.4 and at the end of the twelve months 15,924.0 ± 927.1 and 16,884 ± 882.3 for females and males, respectively. There were statistically significant differences between the different periods.

All patients in Group A accepted the use of the WAT (Question 1); they supported that the WAT helped them increase participation in the ET program and, in general, their physical activity (score 8.1 ± 0.6 in Question 2), and they believed that the WAT increased the adherence to the exercise training program (score 7.35 ± 0.3 in Question 3).

In group A, the 12-month WAT-based ET led to a significant improvement of patients’ functional capacity (by 20%, *p* < 0.05), as estimated by the significant increase by 84.3 ± 9.6 m in the distance walked in the 6-MWT. In addition, a significant linear correlation was observed between the number of steps recorded and the 6-MWT results both at baseline (r = 0.827, *p* < 0.01) and the end of the study (r = 0.807, *p* < 0.01) in group A. From the echocardiographic studies at baseline, all patients had a normal systolic and diastolic function of LV and normal dimensions of LA and LV (Table 2). In addition, no baseline differences were found between the two groups. After one year, the dimensions of LV and LA, as well as the systolic and diastolic function of LV, remained unchanged in both groups.

LVIVSd: left ventricular interventricular septum thickness at end diastole, LVEDD: left ventricular end-diastolic dimension, LVPWd: left ventricular posterior wall thickness at end diastole, LVEDV: left ventricular end-diastolic volume, LVEF: left ventricular ejection fraction, MVE: early diastolic transmitral flow velocity, MVA: late diastolic transmitral flow velocity, MVE/A: ratio of early to late diastolic transmitral flow velocity, LA: left atrium

Similarly, the dimensions of RV and RA at baseline were within the normal range. The systolic function of RV, as measured by TAPSE and TV S’, was also within the normal limits. Still, the RV strain measurements (RVFWSL and RV4CSL) were at lower limits, probably reflecting subclinical changes-abnormalities of RV. The mean PASP was within the normal limits (Table 3). No baseline differences were found between the two groups.

RV bas: right ventricular basal diameter, RVOT prox: proximal right ventricular outflow track diameter from PLAX, RAVol: right atrial volume, TVE: early diastolic transtricuspid flow velocity, TVA = late diastolic transtricuspid flow velocity, TVE/A: ratio of early to late diastolic transtricuspid flow velocity, TVS’: systolic tricuspid annular velocity, TVE’: early diastolic tricuspid annular velocity, TVA’: late diastolic tricuspid annular velocity, TAPSE: tricuspid annular plane systolic excursion, PASP: pulmonary artery systolic pressure, RVFWSL: right ventricular free-wall longitudinal strain, RV4CSL: right ventricular four–chamber longitudinal strain

At the end of the ET program in Group A, TVA decreased by 17% (*p* < 0.05) and TVE/A increased by 13.2% (*p* < 0.05). Moreover, TAPSE, TVS’, RVFWSL and RV4CSL increased by 8.3% (*p* < 0.05), 9.0% (*p* < 0.05), 13.7% (*p* < 0.05) and 26.7% (*p* < 0.05) respectively (Figure 2), while PASP decreased by 7.6% (*p* < 0.05).

No change was found at the end of the study in Group B over time. However, at the end of the study, patients of group A demonstrated increased TAPSE, RVFWSL, and RV4CSL by 6.3% (*p* < 0.05), 11.3% (*p* < 0.05), and 20.3% (*p* < 0.05), respectively, compared to group B.

At the end of the study, there was a significant linear correlation between the number of steps per day recorded and the PASP (r = −0.727, *p* < 0.01), TAPSE (r = 0.831, *p* < 0.001), RV4CSL (r = −0.727, *p* < 0.001) and RVFWLS (r = −0.854, *p* < 0.001) in group A.

## 4. Discussion

The present study results showed that using a WAT during the ET program in patients with CF significantly helped them monitor their physical activity levels, follow the prescribed ET schedule, improve their exercise adherence, and, thus, increase their performance. In addition, a strength of the study was that a significant increase in the number of steps performed by patients during ΕΤ was associated with more benefits on functional capacity and RV cardiac function. Patients with CF were characterized by exercise intolerance, which is multifactorial including age, sex, sedentary lifestyle, nutritional status, the severity of the disease, lung, and heart dysfunction, genotype, and chronic colonization by Pseudomonas aeruginosa, etc. [1,3,12]. Central and peripheral changes decrease the patients’ cardiorespiratory efficiency. Specifically, there is a disease-related decline in pulmonary and cardiac function and a reduced capacity of skeletal-muscle mitochondria to extract/use O_2_. [13,14,15]. The consequent exercise intolerance reduces the quality of life and contributes to increased morbidity and mortality [16]. Excessively concentrated respiratory secretions lead to chronic airway obstruction, combined with the colonization by bacteria, and acute and chronic infections reduce lung function, causing an inability to maintain normal arterial oxygenation and eventually respiratory failure [17]. It was supported that chronic hypoxemia, progressive destruction of pulmonary vascular structure, and persistent increase in cardiac output myocardial are the main reasons for RV hypertrophy and dilatation, the most frequent remodeling in CF, causing cor pulmonale [18,19,20]. The increased pulmonary pressure due to increased stiffness in the pulmonary circulation leads to pulmonary artery hypertension [21]. Moreover, LV, left and right atrium dilatation are usually present in CF patients [22]. A cystic fibrosis-related cardiomyopathy concept also exists [15,18,23]. Thus, impaired cardiac function, causing decreased perfusion to exercise muscle, often exists, leading to skeletal muscle dysfunction and exercise intolerance in CF patients. Additionally, chronotropic incompetence is crucial in limiting exercise in patients with CF [24]. Finally, it was described that cardiac ANS could be altered in CF, causing heart rate variability (HRV) abnormalities [25]. The echocardiographic findings of our patients showed no underlying LV systolic and diastolic and RV systolic function abnormalities, except for marginal results of RV systolic function. Specifically, RV strain measurements (RVFWSL and RV4CSL) were at lower limits, probably due to a subclinical diastolic disturbance. The absence of clinical signs of cardiac dysfunction may be attributed to the fact that only mild cases participated in our study. Ionescu et al. supported that patients with CF have subclinical RV systolic and diastolic dysfunction as measured by tissue doppler echocardiography, even when clinically stable and free from signs of heart failure. The severity of the impairment parallels the severity of the lung disease and the inflammatory process [26]. Moreover, PASP was within the normal limits in our patients.

Regular exercise is recommended to be included in the CF treatment. Several studies have supported that ET is an essential therapeutic intervention in CF to maintain respiratory function, physical fitness, and health-related quality of life [1,2,3]. Systematic reviews and meta-analyses have shown a link between VO_2_ peak improvement and survival in patients with CF [16]. We found a significant improvement in the functional capacity after the 12-month ET program, as assessed by the 6MWT. Specifically, at the end of the 12-month study, there was a substantial increase by 20% in the distance covered by the exercise patients, unlike patients in the control group who showed a non-significant decrease. Notably, the results from the 6MWT were significantly correlated with the number of steps counted by the WAT at the end of our study. A review of the available walking tests concluded that the 6MWT is easy to administer, is better tolerated, and is more reflective of the activities of daily living than the other walking tests [27]. The 6MWT is widely used to assess functional exercise capacity in patients with idiopathic pulmonary fibrosis [28].

Apart from the widely acknowledged cardiovascular and musculoskeletal benefits of exercise in chronic disease patients, studies have described specific mechanisms of physical training to maintain respiratory function [29,30]. It was reported that a single bout of moderate-intensity exercise might have an acute bronchodilator effect, increasing both FEV1 and FEF27-75 [31]. Moreover, another study found that a single bout of maximal exercise improved the lung clearance index in CF [32]. It was suggested that exercise might increase the ease of expectoration and volume of airway secretions, leading to airway clearance [1,33]. The effects of ET on the respiratory function of our patients were not included in this study.

There has been no study examining the effects of ET on the cardiac function in CF patients in the literature. However, the results of our study indicated that the WAT-based ET program did not affect LV anatomy and function. Similarly, no changes were found in LA, RA, and RV anatomy. In contrast, speckle tracking echocardiography revealed an exercise-related subclinical improvement in systolic RV function. Notably, the PASP, TAPSE, RVFWLS, and RV4CSL were correlated with the number of steps counted at the end of the study in Group A. Similar evidence of the dose–response relationship between the exercise prescription and the improvement in QoL, aerobic capacity, and cardiac function in chronic cardiac and pulmonary disease patients has previously been reported [34,35].

Cardiac exercise rehabilitation attenuates cardiac remodeling in cardiovascular disease states, typically defined as mild LV end-diastolic or end-systolic volume changes and improvements in systolic and diastolic cardiac function [36]. Such exercise-induced enhancements in cardiac function are possibly explained by changes in vascular tone and the direct cardiac effects [37,38,39,40]. However, it remains unknown if the exact beneficial mechanisms of exercise also apply to LV dysfunction in CF patients. Interestingly, the study of Estévez-González et al. has shown that an 8-week resistance-training program improved the sympathovagal balance, leading to an increase in HRV [25].

It was demonstrated that exercise primarily affects the RV, which is critical to circulatory function during a performance [41]. La Gerche et al. supported that even in healthy individuals with normal pulmonary vascular function, the hemodynamic load on RV increased relatively more during exercise than that of the LV [42]. The RV response to exercise abnormally in most patients with pulmonary arterial hypertension. Moreover, the increased RV afterload during exercise increases the RV wall stress, leading to long-term clinical deterioration [43]. Therefore, an ET program should be used with caution in patients with severe RV dysfunction. There has been no study about the effects of exercise training on RV dysfunction in patients with CF. Our results indicated that a 1-year ET led to a significant improvement in the RV systolic and diastolic function in our CF patients, who did not have any sign of RV dysfunction.

Some controversies exist about the optimal dose of exercise in CF patients [4]. Guidelines recommend that healthy children and adolescents achieve 60 min of moderate to vigorous physical activity daily, while adults should perform at least 150–300 min of moderate-intensity or 75–150 min of vigorous aerobic physical activity per week [44]. Specifically, the recommendation to walk up to 10,000 steps/day is widely promoted among sedentary populations [45]. ET prescriptions should include aerobic (e.g., walking/jogging or cycling) and anaerobic (e.g., resistance, bodyweight) training. The broad exercise training recommendations above for sedentary people also apply to CF patients [1,2,3,4]. Our training program consisted mainly of aerobic exercise sessions. Τhe aerobic exercise program was performed by walking at least five times per week. The goal was for patients to reach at least the recommended 10,000 steps per day during the first three months of ET and gradually increase the total amount of physical activity. Only the number of steps during their daily physical activity was measured but not its duration Two difficulties that are often encountered in the rehabilitation programs of CF patients are their compliance and maintenance for a sufficient period [3,7,8]. In addition, it is essential to monitor the amount of physical activity they perform correctly so that the training intervention is effective and safe [7,8]. Therefore, in our study, we evaluated the effectiveness and adherence to the program, the recommended progressive changes, and the monitoring for side effects. Close personal contact and individualized counseling helped significantly in our study to promote the patients’ participation in CF training programs. Moreover, the patients accepted the use of WAT as a beneficial method to estimate their physical activity and improve their efforts and adherence. The G1 WAT, which we used, has been chosen in several studies for its high validity in measuring daily step counts in ET and free-living conditions. In addition, it can determine the heart rate during exercise and the energy expended. However, its reliability for such measurements is considered limited [46]. Activity trackers such as pedometers, wrist accelerometers, etc., provide CF patient users with an easy way to objectively monitor their physical activity dimensions (energy expenditure, step count, time spent in physical activity in different intensities, time spent sedentary), and to improve their long-term adherence to adequate daily physical activity [7,8,9]. Finally, WAT helps clinicians to prescribe exercise and physical activity as an integral part of CF management and improve the health outcomes for these patients [7].

There were some limitations to our study. First, we did not include a sedentary healthy control group to compare the physical capacity and cardiac function with the findings of our patients at baseline. However, we used the normal values reported in the literature for these ages to estimate the cardiac function of our patients [11,47]. Second, we used only the 6MWT to assess their physical function. However, the literature supports the reliability of the 6MWT in determining the functional capacity of patients with cystic fibrosis or chronic lung disease [28,48,49,50]. Finally, although we used the WAT in our study to successfully implement the exercise program in patients with CF, the WATs were not applied in the control group to monitor the number of their everyday life steps. This limitation weakens the view that the WAT is a valuable tool to evaluate the functional effects of an ET program in CF patients. Other drawbacks were the limited sample size (mainly because a significant number did not meet the study eligibility criteria, and some patients dropped out from the study) and the fact that they were CF patients with mild disease.

## 5. Conclusions

In conclusion, WAT is a helpful tool for the reliable implementation of an exercise training program in patients with CF. Furthermore, our findings indicate that higher levels of physical function cause more distinct cardiac adaptations in RV.

## Figures and Tables

**Figure 1 sensors-22-04884-f001:**
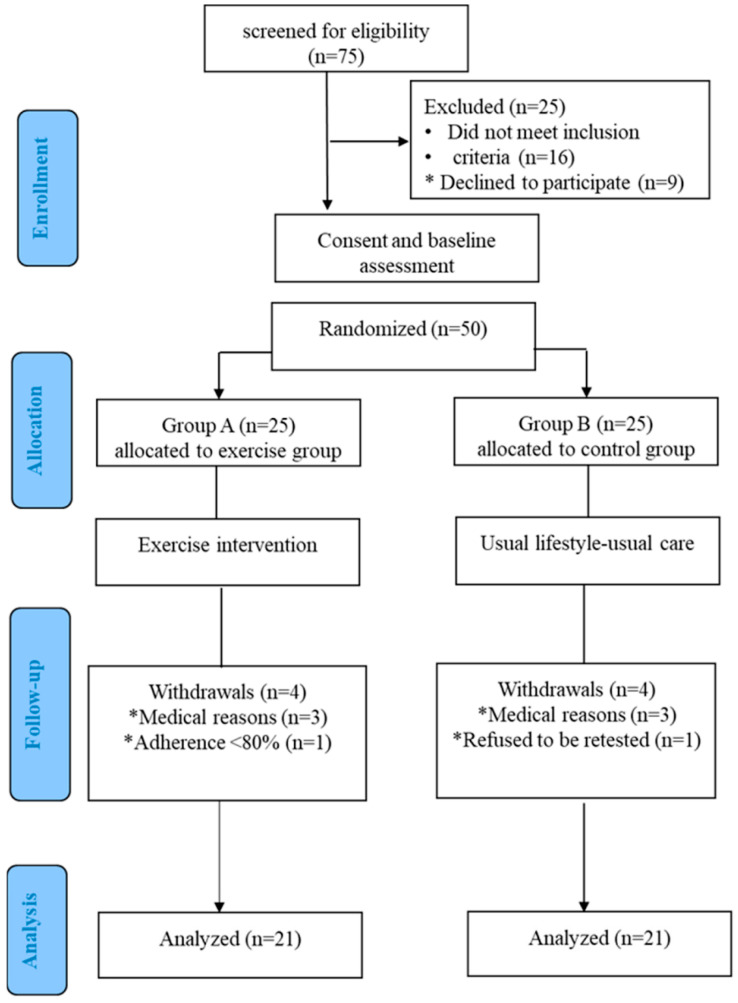
The CONSORT diagram of the study design. * reasons.

**Figure 2 sensors-22-04884-f002:**
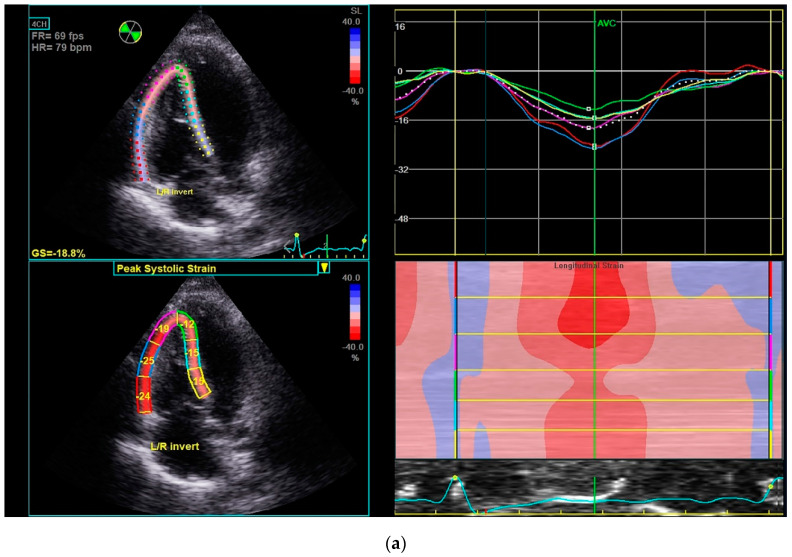
Example of the RV strain measurements in a patient of Group A (**a**) at the baseline (RV4CSL = −18.8%) and (**b**) after the 1-year exercise training program (RV4CSL = −23.9%).

**Table 1 sensors-22-04884-t001:** The demographic and clinical characteristics of the study population at the baseline and after one year ^1^.

Variables	Group A Baseline	Group A After 1 Year	Group B Baseline	Group B After 1 Year
Age (years)	17.0 ± 3.3	18.0 ± 3.3	16.1 ± 3.3	17.1 ± 3.3
BMI (kg/m^2^)	20.0 ± 3.3	20.8 ± 3.5	20.1 ± 2.6	19.7 ± 3.8
BSA (m^2^)	1.50 ± 0.2	1.56 ± 0.1	1.55 ± 0.1	1.56 ± 0.1
SBP (mmHg)	110.8 ± 0.4	118.6 ± 0.9	108.5 ± 0.8	110.04 ± 0.6
DBP (mmHg)	82.2 ± 0.3	81.4 ± 0.4	81.4 ± 0.5	82.1 ± 0.5
HR (bpm)	74.5 ± 5.2	72.4 ± 6.2	76.5 ± 7.1	78.2 ± 6.4

^1^ Data are expressed as mean ± SD. There was no statistically significant difference. BMI: body mass index, BSA: body surface area, SBP: systolic blood pressure, DBP: diastolic blood pressure, HR: heart rate, measured by the electrocardiogram at the beginning and end of the study.

**Table 2 sensors-22-04884-t002:** Echocardiographic data of LV anatomy and function of both groups at baseline and after one year ^1^.

LV Indices	Group A Baseline	Group A After 1 Year	Group B Baseline	Group B After 1 Year
LVIVSd (mm)	7.1 ± 0.7	7.5 ± 0.7	7.5 ± 0.8	7.9 ± 0.8
LVEDD (mm)	43.7 ± 3.5	45.3 ± 4.0	45.0 ± 2.7	46.7 ± 3.8
LVPWd (mm)	6.5 ± 0.6	6.9 ± 0.4	7.0 ± 0.8	7.6 ± 0.9
LVEDV (mL)	91.3 ± 16.3	96.9 ± 17.0	97.5 ± 10.0	93.0 ± 11.5
LVEF (%)	64.4 ± 4.1	66.3 ± 5.2	64.2 ± 4.1	65.3 ± 3.2
MVE (m/s)	0.92 ± 0.1	0.88 ± 0.1	0.90 ± 0.1	0.99 ± 0.1
MVA (m/s)	0.56 ± 0.1	0.54 ± 0.1	0.57 ± 0.1	0.63 ± 0.1
MVE/A	1.7 ± 0.3	1.6 ± 0.3	1.6 ± 0.3	1.6 ± 0.2
LA (cm)	3.1 ± 0.3	3.1 ± 0.2	3.09 ± 0.3	3.13 ± 0.3

^1^ Data are expressed as mean ± SD.

**Table 3 sensors-22-04884-t003:** The echocardiographic data of the RV anatomy and function of both groups at the baseline and after one year ^1^.

RV Indices	Group A Baseline	Group A After 1 Year	Group B Baseline	Group B After 1 Year
RV bas (mm)	37.0 ± 3.0	37.3 ± 4.1	35.3 ± 2.6	35.3 ± 3.3
RV bas/BSA (mm)	25.5 ± 3.7	24.3 ± 3.3	23.8 ± 2.2	22.7 ± 2.5
RVOT prox (mm)	27.9 ± 3.5	27.8 ± 4.6	26.9 ± 2.9	27.4 ± 1.7
RVOTprox/BSA (mm)	19.1 ± 2.7	18.0 ± 2.3	17.6 ± 1.7	17.6 ± 1.6
RAVol/BSA (mL/m^2^)	16.0 ± 2.9	17.4 ± 4.1	15.8 ± 1.6	15.7 ± 1.9
TVE (m/s)	0.75 ± 0.1	0.70 ± 0.10	0.70 ± 0.1	0.69 ± 0.06
TVA (m/s)	0.53 ± 0.1	0.44 ± 0.09 *	0.48 ± 0.1	0.46 ± 0.04
TVE/A	1.4 ± 0.2	1.6 ± 0.2 *	1.5 ± 0.2	1.5 ± 0.1
TVS’ (m/s)	0.11 ± 0.01	0.12 ± 0.01 *	0.11 ± 0.01	0.11 ± 0.01
TVE’ (m/s)	0.13 ± 0.03	0.12 ± 0.01	0.13 ± 0.02	0.12 ± 0.01
TVA’ (m/s)	0.08 ± 0.02	0.09 ± 0.01	0.08 ± 0.01	0.08 ± 0.00
TAPSE (mm)	20.4 ± 2.7	22.1 ± 1.5 *	20.2 ± 1.4	20.8 ± 1.3 ^#^
PASP (mmHg)	24.8 ± 3.5	22.9 ± 3.3 *	24.0 ± 4.0	24.5 ± 3.4
RVFWSL (%)	−22.6 ± 1.4	−25.7 ± 2.5 *	−23.6 ± 2.0	−23.1 ± 1.5 ^#^
RV4CSL (%)	−18.8 ± 1.2	−23.7 ± 2.0 *	−19.1 ± 1.4	−19.7 ± 1.4 ^#^

^1^ Data are expressed as mean ± SD. * *p* < 0.05, group A before and after intervention; ^#^ *p* < 0.05, group A vs. group B after 1-year.

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
