# Peer review of "Effects of a Long-Term Wearable Activity Tracker-Based Exercise Intervention on Cardiac Morphology and Function of Patients with Cystic Fibrosis"

_sensors, 2022, doi:10.3390/s22134884_

Round 1
Reviewer 1 Report
This work investigates the effects of long-term Wearable Activity Tracker-based Exercise 2
Intervention on cardiac morphology and function of Patients 3 with Cystic Fibrosis. In general, this paper is clearly-structured and well-written. But there are a few points the authors may need to consider.
1) A very high proportion, 1/3, of the participants were excluded. The authors are encouraged to provide more details on the reason for them not being included (e.g., numbers that fall into each category of exclusion criteria). This may also need to be discussed and mentioned as a limitation if applied in the real-word monitoring.
2) The authors mentioned that the participants were encouraged to perform any recreational activity. Is there a record or daily how much they did that or these were also considered in the total daily steps? Was this recommendation also included in Group B?
3) The authors wrote 'Moreover, total step counts were computed by summing each exercise training step and calculating median values from the ET days for each participant.' This seems a bit confusing. Did they record the timing of each exercise training.
4) In Table 1, SBP in group A showed an increase. The authors need to explain more about this change.
5) In Table 1, how was heart rate measured?
6) Did the authors perform multiple testing corrections, given that quite a lot of statistical tests were done.
Author Response
Point to point response to:
Reviewer 1
This work investigates the effects of long-term Wearable Activity Tracker-based Exercise Intervention on cardiac morphology and function of Patients with Cystic Fibrosis. In general, this paper is clearly-structured and well-written. But there are a few points the authors may need to consider.
1) A very high proportion, 1/3, of the participants were excluded. The authors are encouraged to provide more details on the reason for them not being included (e.g., numbers that fall into each category of exclusion criteria). This may also need to be discussed and mentioned as a limitation if applied in the real-word monitoring.
We would like to thank the first reviewer for his contribution to the improvement of our article. We have added:
Finally, out of the 75 patients tested for eligibility for the study, 9 eventually refused to participate (7 due to long distance stays and both due to occupational employment) and the remaining 16 due to the presence of non-eligible criteria: active manifestation respiratory infection (6 patients), findings of pulmonary hypertension (4 patients), co-existing cardiac problems (2 patients) and their regular participation in regular exercise programs (4 patients) (lines 72-77)
We have also added in the limitations: ... (mainly because a significant number did not meet the study eligibility criteria, and some patients dropped out from the study) – lines 421-423
2) The authors mentioned that the participants were encouraged to perform any recreational activity. Is there a record or daily how much they did that or these were also considered in the total daily steps? Was this recommendation also included in Group B?
Thank you for this remark.
We have added the following: Their steps during these activities, where the sport allowed it, were counted in the number of total steps (line 152-154)
and Group B: Patients in the control group were asked to continue their usual lifestyle, to participate in recreational physical activities, but refrain from any structured exercise intervention (lines 92, 93)
3) The authors wrote 'Moreover, total step counts were computed by summing each exercise training step and calculating median values from the ET days for each participant.' This seems a bit confusing. Did they record the timing of each exercise training.
Thank you. This part is further clarified in lines 161-163: So, patients were asked to note the number of steps during their daily physical activity and not its duration and in lines 385,386: Only the number of steps during their daily physical activity was measured but not its duration.
4) In Table 1, SBP in group A showed an increase. The authors need to explain more about this change.
This change, as added to the table, was not statistically significant and is probably attributed to the fact that 3 of the patients had a systolic pressure of 120-140 mmHg on the second measurement without any other clinical disorder. (please see table 1, line 204)
5) In Table 1, how was heart rate measured?
Heart rate was measured by resting ECG at the beginning and end of the study in all patients, as added to the table. (please see table 1, line 206)
6) Did the authors perform multiple testing corrections, given that quite a lot of statistical tests were done.
Thank for this remark. We have added the following in line 189: with a Bonferroni correction (Bonferroni post hoc test)
Reviewer 2 Report
Dear Editor,
I had the pleasure to review a well-written study on the effect of a structured physical exercise program base on utilization of a wearable tracker in (almost adult/young adult) patients with cystic fibrosis. While the study is straight-forward some issues still need to be clarified.
In the methods section authors claim that “Patients in the control group were asked 86 to continue their usual lifestyle and refrain from exercise intervention”. This notion poses an ethical issue (since according to guidelines exercise is suggested in this patient population; did control patients provided with an inform consent accordingly?) and a methodological one regarding the usual lifestyle exercise level (which was not measured or reported; as authors note on limitations’ section). Authors are invited to clarify on this (except if by refraining from exercise intervention intend to mean only structured programs).
In the Disuccsion section the authors claim that “In addition, a novelty of the study was the finding that using a WAT, a more significant number of steps was associated with more benefits on functional capacity and cardiac function “. In my opinion, since the study did not compare an exercise program WITH vs WITHOUT WAT monitoring, the notion is not supported by study design and findings and the sentence should be restructured so that no misunderstanding occurs. It is not the WAT being tested in the present study but exercise per se (the control group was asked to refrain from exercise). In the same line the notion in the conclusion section “WAT is a valuable tool for evaluating the functional effects of an ET 388 program in CF” is not supported.
As a general comment I would recommend that Discussion section should focus more on discussing the individual findings of the present study and not reviewing available literature in general.
Author Response
Point to point response to:
Reviewer 2:
I had the pleasure to review a well-written study on the effect of a structured physical exercise program base on utilization of a wearable tracker in (almost adult/young adult) patients with cystic fibrosis. While the study is straight-forward some issues still need to be clarified.
In the methods section authors claim that “Patients in the control group were asked to continue their usual lifestyle and refrain from exercise intervention”. This notion poses an ethical issue (since according to guidelines exercise is suggested in this patient population; did control patients provided with an inform consent accordingly?) and a methodological one regarding the usual lifestyle exercise level (which was not measured or reported; as authors note on limitations’ section). Authors are invited to clarify on this (except if by refraining from exercise intervention intend to mean only structured programs).
We would like to thank the second reviewer for his contribution to the substantial improvement of our manuscript.
Both patients and their parents were informed from baseline that according to the randomization process, there was a 50% probability not to be included in the exercise training group and they should refrain from any structured exercise intervention. This is mentioned in lines 78 and 79: “Patients fulfilling the inclusion criteria were informed about the aim and methodology of the research protocol and provided informed consent.” Moreover, to clarify it, we have added the following part: Patients in the control group were asked to continue their usual lifestyle, to participate in recreational physical activities, but refrain from any structured exercise intervention. (lines 92-93).
In the Discussion section the authors claim that “In addition, a novelty of the study was the finding that using a WAT, a more significant number of steps was associated with more benefits on functional capacity and cardiac function “. In my opinion, since the study did not compare an exercise program WITH vs WITHOUT WAT monitoring, the notion is not supported by study design and findings and the sentence should be restructured so that no misunderstanding occurs. It is not the WAT being tested in the present study but exercise per se (the control group was asked to refrain from exercise). In the same line the notion in the conclusion section “WAT is a valuable tool for evaluating the functional effects of an ET program in CF” is not supported.
Thank you for the remark. We have added the following sentences in the limitations:
Finally, although we used the WAT in our study to successfully implement the exercise program in patients with CF, the WATs were not applied in the control group to monitor the number of their everyday life steps. This limitation weakens the view that the WAT is a valuable tool for evaluating the functional effects of an ET program in CF patients (lines 415-419).
Moreover, we have changed the conclusion accordingly.
In conclusion, WAT is a valuable helpful tool for the reliable implementation of an exercise training program in patients with CF (lines 429-430).
As a general comment I would recommend that Discussion section should focus more on discussing the individual findings of the present study and not reviewing available literature in general.
We have revised the discussion section trying to follow the first reviewer’s recommendation (all changes are marked in red). We would like to thank the first reviewer for his kindness and for his significant contribution to the amendment of the initial submission.
Reviewer 3 Report
The authors presents research to study the effects of wearable activity tracker-based exercise intervention on cardiac morphology and function of patients with Cystic Fibrosis. Overall this manuscript is well written. The project background is introduced with sufficient information. Methods look reasonable and the experiment results look promising. Small suggestion: There are two Figure 2(s), but only 1 figure is presented. In the Figure 2, it would be better if the authors could clearly show the units of the plots.
Author Response
Point to point response to:
Reviewer 3 (Minor):
The authors present research to study the effects of wearable activity tracker-based exercise intervention on cardiac morphology and function of patients with Cystic Fibrosis. Overall this manuscript is well written. The project background is introduced with sufficient information. Methods look reasonable and the experiment results look promising. Small suggestion: There are two Figure 2(s), but only 1 figure is presented. In the Figure 2, it would be better if the authors could clearly show the units of the plots.
Thank you for the remark. Both figures are presented in the result section. Figure 1 refers to the study flowchart and is presented in line 195, while Figure 2 shows an example of the RV strain measurements and is presented in line 257. Moreover, we have added the results of the global strain measurements in the legend of the figure (lines 272-3). Although the units of the plots are presented in the two figures, these are minimized, and it is not easy to see the numbers. We apologize for that.
Round 2
Reviewer 1 Report
All comments have been addressed.